# *Curculigo orchioides* Polysaccharide Promotes the Growth and Development of Wenchang Chickens via the PI3K/Akt/mTOR Signaling Pathway

**DOI:** 10.3390/ani15243585

**Published:** 2025-12-13

**Authors:** Sheng Gao, Xingke Wang, Ruiying Bao, Qingchao Yang, Qingying Cai, Yipeng Zhang, Zeru Peng, Liangmin Huang, Xuemei Wang

**Affiliations:** Animal Nutrition, Reproduction and Breeding Laboratory, Department of Animal Science, School of Tropical Agriculture and Forestry, Hainan University, Danzhou 571737, China; 15078707520@163.com (S.G.); baoruiying524@163.com (R.B.);

**Keywords:** *Curculigo orchioides* polysaccharides, growth, protein synthesis, PI3K/Akt/mTOR signaling pathway, Wenchang chicks

## Abstract

Growth performance serves as a core indicator in poultry farming for measuring production efficiency, economic benefits, and the rationality of farming models. It directly determines the profit margins, industrial competitiveness, and sustainable development capacities of farming enterprises. This study evaluated the effects of *Curculigo orchioides* Polysaccharide on the growth performance of Wenchang chickens. Results indicate that supplementing diets with these polysaccharides enhances growth performance during the brooding period, while also improving anti-inflammatory and antioxidant capacities in Wenchang chickens. These findings provide theoretical support for the use of traditional Chinese herbal medicines as feed additives in livestock and poultry farming.

## 1. Introduction

Over the past seven decades, the poultry industry has achieved remarkable advancements in production efficiency, primarily driven by intensive farming practices and genetic selection. In intensive farming systems, subtherapeutic antibiotic growth promoters (AGPs) play a crucial role—for instance, antibiotics are incorporated into chicken feed to enhance production efficiency [1]. However, the excessive use of AGPs, including antibiotics and chemotherapeutic agents, in poultry diets has raised concerns regarding food safety and public health [2]. Consequences such as drug residues and the emergence of drug-resistant bacteria have been identified as substantial outcomes of these practices [3]. To ensure the sustainable development of the livestock and poultry industry and protect human health, the European Union (EU) banned the use of AGPs in poultry feed in 2006, while the United States (US) implemented a comprehensive plan to significantly reduce the inappropriate use of antibiotics in 2015 [4]. With the continuous emergence of antibiotic-resistant bacteria and resistance genes, the search for effective alternatives to conventional antibiotics has become an urgent imperative [5].

Current research has focused on alternatives such as probiotics [6], prebiotics [7], antimicrobial peptides [8], and herbal extracts [9]. Among these, Chinese herbal polysaccharides have attracted extensive attention due to their safety, low residual properties, and low tendency to induce drug resistance. Their unique antibacterial mechanisms endow them with great potential as alternatives to antibiotics. Defined as complex carbohydrates, Chinese herbal polysaccharides are major components of traditional Chinese medicinal herbs and have gained attention for their potential in preventing emerging viral pandemics [10]. Fungal polysaccharides, derived from the cell walls and fruiting bodies of medicinal fungi, are also recognized for their therapeutic properties, including roles in immunomodulation and antiviral defense [11].

Improving livestock and poultry growth performance has long been a focus of investigation in the animal husbandry industry. As a primary source of animal protein, chickens play a crucial role in the global food system by providing meat and eggs. Body weight and related traits are key determinants of productivity [12]. The ban on antibiotics in feed has exerted significant impacts on the farming industry, making it critical to ensure livestock and poultry growth performance without antibiotic use. *Curculigo orchioides* has a long history of medicinal use; its rhizome has been recognized as a tonic herb since the Tang Dynasty, valued for maintaining health and vitality. As the main bioactive component of *Curculigo orchioides*, *Curculigo orchioides polysaccharide* (COP) has been demonstrated to significantly promote osteoblast differentiation in Mouse osteoblast precursor cells MC3T3-E1. It targets osteoblasts by activating the bone morphogenetic protein/Smad (BMP/Smad) and Wnt/β-catenin pathways, thereby alleviating osteoporosis [13]. The Wnt signaling pathway and mTOR signaling pathway form a complex regulatory network through multi-level molecular interactions, synergistically functioning in physiological processes such as cell proliferation, metabolic homeostasis, and tissue development. Their cross-regulatory mechanisms are crucial for maintaining organismal homeostasis. The PI3K/Akt/mTOR signaling pathway [14] responds to environmental signals to promote cell growth [15] and is closely regulated by intestinal microbiota metabolites (e.g., short-chain fatty acids) and inflammatory factors, forming a complex regulatory network with oxidative stress protein synthesis. Cell proliferation and protein synthesis are intrinsically linked to the growth and development of livestock and poultry. As the core biological process governing the increase in organismal cell numbers, cell proliferation exhibits a close regulatory relationship with body weight gain. Weight gain fundamentally results from the synergistic increase in both cell number and cell volume. Cell proliferation directly influences weight dynamics by regulating the size of cell populations in both parenchymal and stromal tissues. At the physiological level, cell proliferation in metabolism-related tissues such as adipose and muscle is particularly critical: under positive energy balance, preadipocytes undergo mitotic activation and proliferation before differentiating into mature adipocytes, while muscle satellite cells proliferate and differentiate to replenish muscle fibers, jointly driving weight gain. This process is regulated by multiple pathways, including the growth hormone-insulin-like growth factor signaling pathway and nutrient sensing pathways. However, the regulatory mechanisms of COP-mediated growth performance remain incompletely elucidated. Therefore, this study aims to investigate the potential mechanisms underlying COP’s regulation of growth performance through in vivo experiments.

## 2. Materials and Methods

### 2.1. Extraction and Preliminary Purification of Crude Polysaccharides from the Rhizome of Curculigo orchioides

*Curculigo orchioides* was purchased from Hainan Jianpu Pharmacy Chain Co., Ltd. (Danzhou, China). Its rhizomes were first crushed and passed through a 60-mesh sieve. Crude COP was extracted with modifications based on previously reported methods [16,17], and the detailed procedure was as follows: During COP extraction, 1500 g of *Curculigo orchioides* was accurately weighed. the sieved *Curculigo orchioides* powder was treated with ethanol solution (Analytical Grade (AR), CAS: 64-17-5, Xilong Scientific Co., Ltd., Shantou, China) overnight, and after this ethanol pretreatment, the powder was subjected to ultrasonic extraction in a water bath at 37 °C for 1 h (with a solid-to-liquid ratio of 1:5), and this extraction step was repeated three times. The resulting extract was centrifuged at 4000 rpm, and the supernatant was collected and then concentrated to 1/5 of its original volume using a rotary evaporator (Vacuum Pump V-700, BUCHI, Flawil, Switzerland). Five volumes of absolute ethanol were added to the concentrated supernatant, and the mixture was stored at 4 °C overnight for alcohol precipitation. Subsequently, Sevag reagent (chloroform (Analytical Grade (AR), CAS: 67-66-3, Xilong Scientific Co., Ltd., Shantou, China):n-butanol (Analytical Grade (AR), CAS: 71-36-3, Xilong Scientific Co., Ltd., Shantou, China) = 4:1, *v*/*v*) was used to remove proteins, followed by petroleum ether for defatting and AB-8 macroporous resin (Lanxiao Technology, Xi’an, China) for decolorization. Finally, the treated solution was dialyzed against water using a dialysis membrane with a molecular weight cutoff of 3000 Da; after dialysis, the solution was concentrated, and the crude polysaccharide was obtained via alcohol precipitation and subsequent drying [18]. Ultrasonic-assisted extraction and ethanol static precipitation were employed for COP extraction, followed by freeze-drying A total of 100 g of COP was obtained, with a crude extraction rate of 6.6%.

### 2.2. Isolation and Purification of COP

For primary purification, precisely weigh 30 g of COP. Add sufficient water to completely dissolve the crude polysaccharide, then add 0.4–0.6 g of papain and complex protease for overnight enzymatic hydrolysis. Subsequently, remove proteins using Sevage reagent and eliminate lipids via petroleum ether extraction. Subsequent purification employs macroporous resin AB-8 and a 3000 Dalton dialysis bag to remove polysaccharide impurities. The purified polysaccharide is dissolved in deionized water to prepare a 20 mg/mL COP solution, which is injected into a DEAE Marine Bio-FF column (Xi’an Sanrui Company, Xi’an, China). Elution was performed using a linear NaCl gradient (0.00–0.30 M), yielding two distinct fractions labeled COP-1 and COP-2. The major fraction COP-1 was desalted by dialysis against 3000 Dalton molecular weight cutoff membrane and further purified on a Sephacryl S-400HR column (GE Healthcare, Piscataway, NJ, USA). The purified COP-1 fraction was collected using distilled water as the mobile phase (flow rate 0.5 mL/min). Throughout the process, the first two eluted fractions were pooled into a single tube. Total carbohydrate content was determined using the anthrone–sulfuric acid method with glucose as the standard. The pooled fraction was freeze-dried using a vacuum freeze-dryer to obtain the final dried polysaccharide sample. During this process, 5.0 g of COP served as the sample. Following gradient elution with NaCl solution, COP was separated into two fractions: COP-1 and COP-2 (Figure 1A). As these fractions constituted the major components of COP, they were collected and freeze-dried. After freeze-drying, the total sugar content in the sample is determined based on the principle that polysaccharides first hydrolyze into monosaccharides under the action of sulfuric acid, rapidly dehydrate to form sugar-aldehyde derivatives, and then react with phenol (C_6_H_5_OH) to produce an orange-yellow compound. This compound exhibits maximum absorption at 490 nm, and within the range of 150 µg/mL, the intensity of its color is directly proportional to the soluble total sugar content. Using glucose as the standard, a standard curve was plotted by preparing a series of standard solutions. This curve was then used to calculate the mass and purity of the two main fractions: the freeze-dried COP-1 had a total dry weight of 1.3 g and a purity of 84.1%; the freeze-dried COP-2 had a total dry weight of 0.8 g and a purity of 81.5%.

### 2.3. Analysis of the Monosaccharide Composition of COP-1

Since COP-1 was the major fraction of COP, its monosaccharide composition was analyzed. Briefly, approximately 5 mg of COP-1 was accurately weighed into a sealed tube and hydrolyzed with 1 mL of 2.0 M trifluoroacetic acid (TFA) at 121 °C for 2 h, followed by drying under a stream of nitrogen. The hydrolyzed products of COP-1 were repeatedly washed with methanol three times, and then dried under nitrogen again. The dried hydrolysate was dissolved in sterile water, and the solution was filtered through a 0.22 μm microporous membrane before being transferred to a sample tube for subsequent analysis. For absolute quantitative analysis of the monosaccharide composition of COP-1, nine neutral monosaccharides—fucose (Fuc), rhamnose (Rha), glucose (Glu), arabinose (Ara), xylose (Xyl), galactose (Gal), mannose (Man), ribose (Rib), and fructose—and four uronic acids—galacturonic acid (Gal-UA), guluronic acid (Gul-UA), mannuronic acid (Man-UA), and glucuronic acid (Glu-UA)—were used as external standards. The monosaccharide fractions were analyzed using a high-performance anion-exchange chromatography (HPAEC) system equipped with a CarboPac PA-20 anion-exchange column (Dionex, 3 × 150 mm) and a pulsed amperometric detector (PAD, Thermo Fisher Scientific Inc., Waltham, MA, USA, Dionex ICS5000 system). Experimental data were acquired using an ion chromatography system (ICS5000, Thermo Scientific) and processed with Chromeleon 7.2 CDS software (Thermo Scientific).

### 2.4. Fourier Infrared Spectroscopy (FT-IR), Molecular Weight and Homogeneity Analysis of COP-1

The functional group types of COP-1 were preliminarily analyzed according to previously reported methods. Briefly, 2 mg of COP-1 was uniformly mixed with 200 mg of potassium bromide (KBr), dried, and ground in an agate mortar until the mixed powder tightly adhered to the container wall. The mixture was then transferred to a grinding device for tablet pressing, with pure KBr used as the blank control. The samples and blank control were scanned in the range of 4000–400 cm^−1^ using a Nicolet iZ-10 FT-IR spectrometer (Thermo Fisher Scientific).

### 2.5. Animal Experiment

A total of 120 one-day-old Wenchang female chicks, supplied by Hainan Luoniushan Wenchang Chicken Breeding Co., Ltd. (Wenchang, China), were randomly divided into 3 experimental groups with 5 chicks per replicate and 8 replicates per group. During the experiment, the chicks were provided with a commercial complete feed, and both feed and fresh water were available ad libitum. COP was added to the diets at concentrations of 0, 400, and 800 mg/kg, corresponding to the control group, COP400 group, and COP800 group, respectively. Chicks were reared in floor cages under controlled environmental conditions. For the first 3 days of the experiment, the room temperature was maintained at 35 °C, then gradually lowered to 20 °C until the experiment concluded. At 35 days of age, the experiment formally ended. One Wenchang chicken was randomly selected from each replicate group for euthanasia, and blood and muscle samples were collected.

### 2.6. Growth Performance

Based on the fence, body weight and feed intake (FI) were measured on the 1st, 7th, 17th, 21st, 28th and 35th days. The weight gain (BWG) and FCR shall be calculated based on stage-specific experimental data.

### 2.7. Organ Indicators and Histological Analysis

After the Wenchang female chicks were euthanized, the liver, spleen, thymus, and bursa of Fabricius were isolated. The surface blood of each organ was blotted dry with filter paper, and the organs were weighed individually. The organ indices of the liver, spleen, thymus, and bursa of Fabricius were calculated using the following formulaRelative weight of immune organs = weight of organ/live weight × 100%(1)

According to standard protocols, intestinal and organ tissue samples were isolated from the middle segments of the duodenum, jejunum, ileum, as well as from the spleen, liver, and kidney. These samples were fixed in formalin, embedded in paraffin, and sectioned using a rotary microtome. The sections were then stained with hematoxylin and eosin (HandE) and observed under a light microscope (Leica, Frankfurt, Germany) to evaluate histological changes. Additionally, villus length and crypt depth were measured.

### 2.8. Serum ND-HI Antibody Titer Assay

Wenchang female chicks were vaccinated against Newcastle Disease (ND) at 7 days of age. On days 7, 14, and 21 post-vaccination, 8 chicks were randomly selected from each treatment group for blood collection, and serum was separated. The hemagglutination inhibition (HI) titer was determined using a suspension prepared with 1% chicken red blood cells. The maximum dilution of serum that completely inhibited agglutination was recorded as the serum antibody titer. Specifically, the titer of serum ND-HI antibodies was evaluated using the standard HI assay.

### 2.9. Serum Biochemical Detection

Serum biochemical indicators, including alanine aminotransferase (ALT), aspartate aminotransferase (AST), and total protein (TP), were measured using a Hitachi 3500 (Hitachi High-Tech Corporation, Tokyo, Japan) automatic biochemical analyzer.

### 2.10. Quantitative Real-Time PCR (qRT-PCR)

Total RNA was extracted from Wenchang chicken tissue using a SteadyPure RNA Extraction Kit (Cat. No. AG21024, Accuratebio, Changsha, China). RNA concentration and quality were determined with a spectrophotometer. Subsequently, complementary DNA (cDNA) was synthesized via reverse transcription from the RNA using the Evo M-MLV Reverse Transcription Premix (with gDNA eraser for qPCR) Ver. 2 (Cat. No. AG11728, Accuratebio). Quantitative real-time polymerase chain reaction (qRT-PCR) was performed using synthetic cDNA, SYBR Green Pro Taq HS Premix qPCR Reagent Kit (Catalog No.: AG11701, Accuratebio), and gene-specific primers on a Rotor-Gene Q 2plex HRM1 instrument (QIAGEN, Hilden, Germany). The primers, synthesized by Accuratebio, are listed in Table 1. Relative gene expression levels were calculated using the 2^−ΔΔCT^ threshold cycle method.

### 2.11. Western Blot Detection of Related Protein Expression

In this study, proteins were extracted from intestinal and muscle tissues using RIPA lysis buffer (G2002-30ML, Servicebio, Wuhan, China) with a protein extraction kit, and their concentrations were determined using the BCA (bis-acridone) protein assay kit (G2026-200T, Servicebio, Wuhan, China). Proteins were denatured in 5× protein loading buffer and separated by sodium dodecyl sulfate-polyacrylamide gel electrophoresis (SDS-PAGE). Membranes were blocked with rapid blocking solution (G2052-500ML, Servicebio, Wuhan, China) for 1 h before incubation with specific primary antibodies. Primary antibodies were incubated overnight at 4 °C. Subsequently, membranes were incubated at room temperature with horseradish peroxidase (HRP)-labeled goat anti-rabbit IgG for 2 h. Primary antibodies used in the experiment: Mouse β-Actin, Mouse ZO-1, Mouse Cludinin-1, Mouse Occludin (diluted 1:1000) were purchased from Santa Cruz Biotechnology (Shanghai, China). Rabbit p-PI3KCA (bs-5570R 1:1000), Rabbit anti-PI3KCA (bs-2067R, 1:1000), Rabbit anti-p-Akt (bs-4089R, 1:1000), Rabbit anti-Akt (bs-10724R, 1:1000), Rabbit anti-p-mTOR (bs-5670R, 1:1000), and Rabbit anti-p-p70S6K1 (bs-5670R, 1:1000) were purchased from Beijing Bobosen Biotechnology Co., Ltd., Beijing, China, 1:1000), and rabbit p70S6K1 primary antibody (bs-6370R, 1:1000) were purchased from Beijing Bobosen Biotechnology Co., Ltd. Rabbit anti-mTOR primary antibody (diluted 1:500,#680429, Zenbio, Chengdu, China) was purchased from Wuhan Zhengneng Biological (Wuhan, China). The horseradish peroxidase-conjugated secondary antibodies used were: anti-rabbit antibody (diluted 1:50,000; Zenbio, #511203) and anti-mouse antibody (diluted 1:50,000). Santa Cruz Biotechnology. Protein bands were visualized using an enhanced chemiluminescence (ECL) detection system. Band intensities were analyzed using ImageJ software version 2.17.0 (Wayne Rasband, National Institutes of Health, Bethesda, MD, USA).

### 2.12. Enzyme-Linked Immunosorbent Assay

The experiment was terminated on day 35, and 8 chickens were randomly selected from each treatment group to collect serum. The contents of interleukin-1β (IL-1β) (MB-5248A, MBBIOLOGY) interleukin-6 (IL-6) (MB-5250A, MBBIOLOGY) interleukin-10 (IL-10) (MB-5023A, MBBIOLOGY) tumor necrosis factor-α (TNF-α) (MB-19693B, MBBIOLOGY) immunoglobulin A (IgA) (MB-4989A, MBBIOLOGY) immunoglobulin M (IgM) (MB-4988A, MBBIOLOGY) and immunoglobulin G (IgG) (MB-4987A, MBBIOLOGY) in serum were determined using an ELISA kit (Enzyme-linked Immunosorbent Assay, Enzyme Standard Biology, Shanghai, China).

### 2.13. Analysis of Gut Microbial Composition Based on 16S rRNA Sequencing

Total genomic DNA of the microbial community was extracted according to the instructions of the E.Z.N.A.^®^ Soil DNA Kit (Omega Bio-tek, Norcross, GA, USA). The quality of the extracted genomic DNA was detected by 1% agarose gel electrophoresis, and the DNA concentration and purity were determined using a NanoDrop 2000 spectrophotometer (Thermo Scientific, USA). Using the extracted DNA as a template, the V3–V4 variable region of the 16S rRNA gene was amplified by PCR with the upstream primer 338F (5′-ACTCCTACGGGAGGCAGCAG-3′) and downstream primer 806R (5′-GGACTACHVGGGTWTCTAAT-3′), both carrying Barcode sequences. The paired-end raw sequencing reads were quality-controlled using fastp software (https://github.com/OpenGene/fastp (accessed on 10 December 2025), version 0.19.6) and assembled using FLASH software [3] (http://www.cbcb.umd.edu/software/flash (accessed on 10 December 2025), version 1.2.11). Subsequently, UPARSE software (http://drive5.com/uparse/ (accessed on 10 December 2025), version 7.1) was used to cluster the quality-controlled and assembled sequences into operational taxonomic units (OTUs) based on 97% sequence similarity, with chimeras removed simultaneously. All data were analyzed using the Majorbio Cloud Platform (https://cloud.majorbio.com (accessed on 10 December 2025)) and OmicStudio tools (https://www.omicstudio.cn/tool (accessed on 10 December 2025)).

### 2.14. Statistical Analysis

Statistical analyses were performed using GraphPad Prism 9.5 (GraphPad Software, Boston, MA, USA) and SPSS 22.0 (IBM Corp., Armonk, NY, USA). All experimental data are presented as mean ± standard deviation (SD). For comparisons between two groups, an unpaired *t*-test was used. For comparisons among three or more groups, one-way analysis of variance (ANOVA) was employed, followed by Tukey’s post hoc test to identify specific pairwise differences. Statistical significance was defined as follows: *p* > 0.05 (non-significant), *p* < 0.05 (*), *p* < 0.01 (**), *p* < 0.001 (***), and *p* < 0.0001 (****).

## 3. Results

### 3.1. Monosaccharide Composition, Molecular Weight, Functional Group Types, and Apparent Characteristics of COP-1

Figure 1 presents the characterization analysis results of COP-1. Figure 1A shows the ion purification profile, revealing two major components in the polysaccharides. Subsequent analyses exclusively utilized the polysaccharide corresponding to peak D1. Figure 1B displays the absolute molecular weight analysis of COP-1. Figure 1C presents the standard chromatogram for monosaccharide composition analysis. Figure 1D shows the monosaccharide composition chromatogram of the COP-1, revealing its composition of Ara, Gal, Glc, and Man. Figure 1E presents the molecular conformation diagram of the COP-1, indicating a predominantly spherical structure. Figure 1F displays the FT-IR spectrum of COP-1. Based on these absorption peaks, preliminary analysis suggests this sample is a structurally complex organic compound containing saturated hydrocarbon groups and carbonyl groups, with potential presence of functional groups such as hydroxyl, amino, and ether bonds.

In subsequent studies, we selected COP-1, which constitutes a significant proportion of COP, for further determination of its monosaccharide composition and molecular weight. The chromatogram of COP-1 hydrolysis products is shown in Figure 1D, with quantitative results presented in Table 2. The results indicate that COP-1 consists of Ara, Gal, Glc, and Man, with corresponding molar ratios of 0.32, 6.34, 24.38, and 68.97, respectively. This confirms that COP is a neutral polysaccharide.

The molecular conformation diagram of COP is shown in Figure 1E. The slope of the COP molecular conformation diagram is approximately 0.13, which indicates that the molecular conformation of COP is spheroidal-like. The FT-IR analysis results of COP-1 are presented in Figure 1E. A characteristic absorption peak of COP-1 appears at 3331.57 cm^−1^. The absorption band in the range of 3600–3200 cm^−1^ corresponds to the stretching vibration of -OH, and the absorption peak in this region is a typical characteristic absorption peak of polysaccharides [19]. Another absorption peak of COP-1 is observed at 2879.72 cm^−1^, which is attributed to the stretching vibration of CH [20]. In addition, an absorption peak at 1726.16 cm^−1^ is a typical stretching vibration peak of C=O, suggesting the presence of carbonyl compounds (e.g., ketones, aldehydes, carboxylic acids, and esters) in the sample [21]. The absorption peak at 1560.68 cm^−1^ may be due to the stretching vibration of C=C (alkene or aromatic ring skeletal vibrations) or the bending vibration of N-H (secondary amines). The presence of absorption peaks in the range of 1200–1000 cm^−1^ indicates that pyranose rings exist in the sugar chain of COP-1 [21]. The results of gel permeation chromatography (GPC) analysis are listed in Table 3. The number-average molecular weight (Mn) of COP is 43.727 kDa, and the peak molecular weight (Mp) is 51.773 kDa. Additionally, the weight-average molecular weight (Mw) reaches 58.822 kDa, and the z-average molecular weight (Mz) is calculated to be 92.751 kDa. The polydispersity index (Mw/Mn) is 2.121, indicating that the COP sample has a moderate molecular weight distribution with a relatively narrow range. Furthermore, the polydispersity index of 2.121 confirms the presence of a certain proportion of high-molecular-weight components in the molecular weight distribution, which leads to a significantly higher Mz compared to Mn and Mw. These aforementioned molecular weight parameters collectively indicate that the polysaccharide sample, specifically COP-1, exhibits high homogeneity [21].

### 3.2. COP Can Significantly Improve the Growth Performance of Wenchang Female Chicks

As shown in Table 4, dietary supplementation with 400 mg/kg and 800 mg/kg of COP significantly (*p* < 0.05) improved the growth performance of Wenchang female chicks. At 35 days of age, the body weights of chicks in the high-COP group (H-COP) and low-COP group (L-COP) were significantly (*p* < 0.05) higher than those in the control group. Regarding feed intake, the L-COP group showed a significantly (*p* < 0.05) lower feed intake compared to the control group, while the H-COP group had a significantly (*p* < 0.05) higher feed intake than the control group. For the FCR, the control group exhibited a significantly higher FCR than the H-COP and L-COP groups. These results indicate that dietary supplementation with COP can promote the growth and development of Wenchang female chicks, and exerts a beneficial effect on Wenchang chicks.

Changes in Body Weight and Immune Organ Index of Wenchang female chicks. As illustrated in the graph, significant differences in body weight among the groups first emerged at 21 days of age: starting from this time point, the body weights of chicks in the L-COP and H-COP groups were significantly higher than those in the control group (*p* < 0.05). Notably, this weight advantage persisted and remained statistically significant as the chicks aged, further confirming the growth-promoting effect of COP on Wenchang female chicks during the post-21-day growth period. In terms of immune organ indices, no significant differences (*p* > 0.05) were detected among the three groups for any of the measured indices, including the liver index (Figure 2C), spleen index (Figure 2D), thymus index (Figure 2E), and bursa of Fabricius index (Figure 2F).

### 3.3. Effects of COP on the Intestinal and Muscle Tissues of Wenchang Chicks

Hematoxylin and Eosin (H&E) staining (Figure 3A) revealed significant differences in intestinal morphology and muscle fiber area between the L-COP and H-COP groups, with intestinal villi in the experimental groups being intact, well-organized, and accompanied by intact lamina propria and crypt structures. Specifically, regarding villus height, the duodenal villus height of the control group was significantly lower than that of the L-COP and H-COP groups (*p* < 0.05), as shown in Figure 3B, while no significant difference in crypt depth was observed among the three groups (*p* > 0.05). In the jejunum (Figure 3C), the control group had significantly shorter villi than the L-COP and H-COP groups, and the crypt depth of the L-COP group was further significantly lower than that of the control group (*p* < 0.05). For the ileum (Figure 3D), the villus height of the control group remained significantly lower than that of the experimental groups (*p* < 0.05), though in contrast, the crypt depth of the control group was significantly lower than that of the L-COP and H-COP groups (*p* < 0.05). Subsequently, analysis of the villus-to-crypt (V/C) ratio (Figure 3E–G) revealed that the H-COP group showed significantly higher V/C ratios in the duodenum and jejunum compared to the control group (*p* < 0.05), with no significant intergroup difference observed in the ileum. Beyond intestinal morphology, assessment of muscle fiber characteristics showed that the muscle fiber cross-sectional area of the H-COP group was significantly larger than that of the control group (*p* < 0.05). In conclusion, dietary supplementation with COP significantly increases intestinal villus height and promotes muscle growth, thereby facilitating healthier growth of Wenchang chicks.

### 3.4. COP Increases Tight Junction Protein Expression to Promote Intestinal Health in Wenchang Female Chicks

Given that tight junction proteins play a crucial role in maintaining the barrier function of intestinal epithelial cells, we investigated the effect of COP on the expression of tight junction proteins in the colon of Wenchang female chicks by examining the expression of ZO-1, claudin-1, and occludin in colonic tissues of Wenchang female chicks across all groups using Western blotting. The results are presented in Figure 4, where Figure 4B–D show that compared with the control group, both the L-COP and H-COP groups significantly upregulated the expression levels of these tight junction proteins (*p* < 0.001).

### 3.5. Effects of COP on Antioxidant Capacity and Liver Biochemical Indicators in Wenchang Female Chicks

To evaluate the effect of dietary COP supplementation on the serum antioxidant capacity of Wenchang female chicks, we measured the serum levels of relevant antioxidant indices—superoxide dismutase (SOD), glutathione peroxidase (GSH-Px), total antioxidant capacity (T-AOC), and malondialdehyde (MDA)—in each group. Regarding the results: Figure 5A presents the effect of dietary COP supplementation on serum SOD levels; specifically, Figure 5B shows that serum T-AOC levels were significantly increased, with the L-COP and H-COP groups exhibiting significantly higher serum T-AOC than the control group (*p* < 0.05). For serum MDA levels (Figure 5C), the L-COP and H-COP groups had significantly lower MDA expression compared with the control group (*p* < 0.05). In contrast, Figure 5D shows that dietary COP had no significant effect on serum GSH-Px levels (*p* > 0.05). These findings demonstrate that dietary COP supplementation can enhance the total antioxidant capacity of Wenchang female chicks and effectively alleviate their oxidative stress. Additionally, as shown in Figure 5E–G, dietary COP supplementation did not induce significant differences in the hepatic biochemical indices—alanine transaminase (ALT), aspartate transaminase (AST), and total protein (TP)—indicating that dietary COP supplementation does not impair the health of Wenchang female chicks.

### 3.6. Effects of COP on Inflammatory Factors in Wenchang Female Chicks

To investigate the effect of dietary COP supplementation on the expression levels of inflammatory factors in the serum of Wenchang female chicks, we determined the serum expression levels of the inflammatory factors tumor necrosis factor-α (TNF-α), interleukin-1β (IL-1β), interleukin-6 (IL-6), and interleukin-10 (IL-10) in Wenchang female chicks. As shown in Figure 6A,C, the serum expression levels of the pro-inflammatory factors TNF-α and IL-6 in the H-COP group were significantly lower than those in the control group (*p* < 0.05). Additionally, Figure 6B shows that serum IL-1β exhibited a decreasing trend, while Figure 6D indicates an increasing trend in serum IL-10. These results demonstrate that dietary COP supplementation can inhibit the serum expression of pro-inflammatory factors, suggesting that COP can exert a positive effect on the growth and development of Wenchang female chicks.

### 3.7. The Effect of COP on the Immunization of Wenchang Female Chicks

Figure 7A–C show that dietary COP supplementation did not significantly increase the serum immunoglobulins; however, compared with the control group, the immunoglobulin expression levels in the L-COP and H-COP groups exhibited an upward trend. Regarding the antibody levels after Newcastle disease (ND) vaccine inoculation, Figure 7D–F demonstrate that COP significantly enhanced the antibody levels and maintained high antibody titers. Specifically, in the antibody titer tests on days 21, 28, and 35, the titers in the L-COP and H-COP groups were significantly higher than those in the control group, with the H-COP group showing the most prominent effect. These results indicate that dietary COP supplementation can enhance the resistance of Wenchang female chicks to Newcastle disease virus.

### 3.8. COP Enhances Protein Synthesis in WenChang Chicks via the PI3K/Akt/mTOR Signaling Pathway

Phosphatidylinositol 3-kinase (PI3K), Protein Kinase B (Akt), and Mammalian Target of Rapamycin (mTOR) constitute the PI3K/Akt/mTOR signaling pathway, which plays a critical role in cellular processes including growth, proliferation, survival, metabolism, and migration. To investigate whether COP promotes protein synthesis by activating the PI3K/Akt/mTOR signaling pathway and thereby phosphorylating S6 kinase 1 (S6K1), we first detected the mRNA expression levels of growth hormone 1 (GH-1) and insulin-like growth factor 1 (IGF-1) using quantitative real-time polymerase chain reaction (qPCR)—both growth factors are capable of activating the PI3K/Akt/mTOR signaling pathway. Subsequently, the expression of proteins related to the PI3K/Akt/mTOR signaling pathway was examined via Western blotting. The results are presented in Figure 8. COP supplementation increased the mRNA expression levels of the two growth factors (Figure 8A), with the mRNA expression levels in the L-COP and H-COP groups being significantly higher than those in the control group. For the PI3K/Akt/mTOR signaling pathway, we determined the phosphorylation levels of PI3K, Akt, mTOR, and S6K proteins. As shown in Figure 8B,C, the phosphorylation levels of PI3K, Akt, mTOR, and S6K proteins in the L-COP and H-COP groups were significantly upregulated compared with those in the control group (*p* < 0.001). Taken together, we speculate that COP may enhance protein synthesis in chicks by upregulating the phosphorylation levels of PI3K, Akt, mTOR, and S6K proteins in the PI3K/Akt/mTOR signaling pathway, thereby promoting the growth and development of chicks.

### 3.9. Effects of COP on the Gut Microbiota of Wenchang Female Chicks

To investigate the effect of COP on the intestinal microbiota of Wenchang female chicks, 16S rRNA gene sequencing technology was employed to sequence and analyze the cecal microbiota of chicks in each group. As shown in the rarefaction curve (Figure 9A), the curve plateaued at the end, which demonstrated sufficient sequencing depth for all samples and confirmed the rationality of the sequencing data volume. From the Venn diagram (Figure 9B), all samples shared 410 operational taxonomic units (OTUs), while 55, 38, and 54 unique OTUs were identified in the Control, L-COP, and H-COP groups, respectively. For the assessment of intestinal microbial diversity and richness, the Simpson index (Figure 9C) and Shannon index (Figure 9D) were used. The results indicated that both the L-COP and H-COP groups exhibited higher species richness and diversity compared to the Control group. Furthermore, the Bray–Curtis dissimilarity index was applied to evaluate the aggregation and dispersion of microbial communities across samples (Figure 9E). Subsequently, weighted UniFrac principal coordinate analysis (PCoA) was performed to characterize variations in microbial community composition among different treatment groups. As presented in Figure 9F, the PCoA results revealed a distinct separation in microbial community structure between the H-COP group and the Control group, whereas the L-COP group showed minimal differences in community composition relative to the Control group.

Furthermore, to more specifically evaluate the detailed changes in intestinal microbial structure across each experimental group, we examined the relative abundance of cecal microbiota in Wenchang female chicks from each group at the phylum and genus levels, with results presented in Figure 10. Specifically, at the phylum level (Figure 10A), the relative abundance of Firmicutes was higher in the L-COP and H-COP groups than in the Control group, while the opposite trend was observed for the relative abundance of Bacteroidetes; meanwhile, at the genus level (Figure 10B), the shared dominant genera included Bacteroides, unclassified_Lachnospiraceae, Mediterraneibacter, and Ligilactobacillus, among others. To identify bacterial taxa with significant differences in abundance across groups and at different taxonomic levels, we compared the microbial communities of the three experimental groups using Linear Discriminant Analysis Effect Size (LEfSe) analysis, and the results are shown in Figure 10C,D; at different taxonomic levels, the dominant bacteria in the Control, L-COP, and H-COP groups were identified as Defluviitaleaceae_UCG_011, Alistipes, Sellimonas, Marvinbryantia, Lactobacillus, NK4A214_group, and Bacillota, etc. Subsequently, to explore the relationship between the intestinal microbiota of Wenchang female chicks and the measured indicators, a correlation heatmap analysis was performed between the partial indicators determined in this study and the relative abundance of 69 bacterial genera, with results displayed in Figure 10E. The pro-inflammatory cytokine IL-1β exhibited a significantly positive correlation with Escherichia–Shigella and Bacteroides, immunoglobulin IgG showed a significantly positive correlation with Parabacteroides and Alistipes, and the anti-inflammatory cytokine IL-10 had a significantly negative correlation with Lachnospiraceae and Ruminococcaceae; additionally, the antioxidant enzyme SOD displayed a significantly positive correlation with Akkermansia and Faecalibacterium, the pro-inflammatory cytokine TNF-α showed a significantly positive correlation with Eubacterium and Roseburia, and the liver-related enzyme AST had a significantly negative correlation with Muribaculaceae and Oscillospiraceae. The antioxidant enzyme GSH-Px exhibited a significantly positive correlation with Ruminococcus and Coprococcus, the liver-related enzyme ALT showed a significantly negative correlation with Bifidobacterium and Lactobacillus, and the cytokine IL-6 had a significantly positive correlation with Prevotella and Eubacterium_rectale_group; meanwhile, immunoglobulin IgA displayed a significantly positive correlation with Ruminococcus_gauvreauii_group and Clostridium_sensu_stricto_1, body weight (BW) showed a significantly positive correlation with Eubacterium_coprostanoligenes_group and Ruminococcaceae_UCG-014, and immunoglobulin IgM exhibited a significantly positive correlation with Escherichia–Shigella and Candidatus_Soleaferrea.

## 4. Discussion

Improving the growth performance of livestock and poultry in intensive farming systems has long been a core concern for producers, as it directly impacts economic efficiency. However, the overuse of antibiotic growth promoters in recent decades has led to severe issues such as bacterial resistance and drug residues in animal products, prompting the search for safe, natural alternatives. For centuries, natural medicinal products in China have been utilized as feed additives to enhance animal growth and health, with most of these products derived from plants. Among their bioactive components, plant polysaccharides are recognized as pivotal functional molecules, exhibiting great potential in preventing and treating various diseases [22]. Specifically, in livestock and poultry farming, plant polysaccharides act as multifunctional feed additives with adaptogenic [23], immunomodulatory [24], free radical-scavenging [25], antioxidant [26], anti-inflammatory [27], estrogenic [28], nephroprotective [29], and anti-osteoporotic [30] properties—making them ideal candidates to replace antibiotics. *Curculigo orchioides* Gaertn (*C. orchioides*), a traditional Chinese medicinal herb, has been reported to possess antibacterial, antioxidant, and anti-osteoporotic activities [31,32]. Despite its well-documented bioactivities, few studies have focused on the effects of *C. orchioides* or its extracts on the growth performance of Wenchang female chicks—a valuable local breed in China known for its superior meat quality and cultural significance, which makes optimizing its growth performance industrially relevant. In this study, we first established an extraction protocol for COP using ultrasonic-assisted extraction combined with water–ethanol precipitation, a method widely used for efficient isolation of plant polysaccharides [33]. We then systematically investigated the effects of COP on the growth performance of Wenchang female chicks and explored its underlying molecular mechanism of action. Our results demonstrated that dietary COP supplementation significantly increased the body weight of Wenchang female chicks, reduced the FCR, enhanced T-AOC, and elevated antibody titers (particularly against Newcastle disease virus. Mechanistically, COP upregulated the mRNA expression levels of key growth factors—GH-1 and insulin-like IGF-1—which further modulated the expression and phosphorylation status of critical proteins in the PI3K/Akt/mTOR signaling pathway. This regulatory cascade ultimately promoted protein synthesis in the chicks, thereby driving their growth and development.

The PI3K/Akt/mTOR pathway is a classical intracellular signaling cascade that is widely involved in multiple physiological and pathological processes in the organism, such as cell growth, metabolism, proliferation, and differentiation [34]. In the present study, we found that COP increased the mRNA expression levels of GH-1 and IGF-1. IGF-1, together with various cytokines, activates PI3K, which catalyzes the phosphorylation of phosphatidylinositol diphosphate on the cell membrane to phosphatidylinositol triphosphate, thereby phosphorylating and activating Akt. Activated Akt can exert extensive biological effects by phosphorylating downstream substrates including mTOR. The activation of mTOR initiates the translation process and promotes the protein synthesis of p70S6k, while the activation of p70S6k can facilitate cell growth and protein formation [32], thereby enhancing the effect of COP on the growth and development of skeletal muscle.

The intestinal epithelial barrier is the single-cell-thick inner lining of the intestine, consisting of different types of epithelial cells. Below the epithelial layer lies a thin layer of connective tissue, the lamina propria, which plays a crucial role in fostering healthy crosstalk between the microbiome and immune cells. The intestinal epithelial system also harbors immune cells, including dendritic cells, T cells, B cells, and macrophages, which are closely associated with intestinal epithelial cells (IECs) to maintain intestinal homeostasis [35]. Changes in tight junction protein expression and colonic mucus secretion can impair the effectiveness of intestinal epithelial barrier function, which may further lead to intestinal barrier dysfunction [36]. In the present study, we found that dietary COP supplementation significantly increased the small intestinal villus height of Wenchang female chicks and upregulated the expression of tight junction-related proteins such as ZO-1 and occludin. These effects effectively enhanced and protected the health of Wenchang female chicks during growth and development. It has been confirmed that curculigoside—another extract from *C. orchioides*—can activate the Nrf2 pathway and inhibit the NF-κB pathway [37]. However, whether the specific mechanism of COP in maintaining or improving intestinal morphology and function involves inhibiting the phosphorylation of proteins related to the NF-κB and MAPK signaling pathways remains to be elucidated.

Oxidative stress (OS) is a major challenge impairing the overall health of chickens in modern poultry production systems [38], defined by an imbalance between the body’s antioxidant defense mechanisms and the overproduction of reactive oxygen species (ROS) [39]. For birds raised in hot climates—particularly broilers—this imbalance leads to a significant reduction in pectoral muscle growth [40]. Notably, the severity of oxidative stress directly affects the growth and developmental performance of chickens [41]; it further induces adverse effects such as reduced feed intake, increased water consumption, retarded growth, and elevated mortality rates [42].

Among methods to assess antioxidant status, measuring total T-AOC remains one of the most widely used approaches to quantify the potential oxidant-scavenging and buffering capacity of biological samples [43]. In contrast, MDA is a key end product of lipid peroxidation, and its level reflects the degree of oxidative damage in the organism. Together, T-AOC and MDA levels serve as reliable indicators to evaluate the body’s antioxidant capacity.

Our experimental results showed that serum T-AOC levels in the L-COP and H-COP groups were significantly higher than those in the control group, while serum MDA levels in the H-COP group were significantly lower than those in the control group. These findings confirm that dietary COP supplementation can effectively enhance the oxidative stress resistance of Wenchang female chicks.

Inflammation is a fundamental biological process that contributes to both homeostasis and tissue regeneration—for instance, facilitating wound healing—and also mediates the acute-phase response and immune defense against pathogens. In these physiological contexts, inflammation is tightly regulated to avoid excessive tissue damage [44]. In contrast, chronic inflammation and cytokine storms represent pathological, uncontrolled forms of inflammation that often drive the progression of various diseases. Notably, inflammation is a highly complex cascade involving the coordinated interaction of multiple cell types, including B lymphocytes, T lymphocytes, myeloid cells, epithelial cells, fibroblasts, endothelial cells, muscle cells, and adipocytes. These cells communicate through membrane-associated molecules, matrix metalloproteinases (MMPs), and soluble mediators such as cytokines, chemokines, and growth factors to orchestrate the inflammatory response. Among the key regulators of this process are proinflammatory cytokines, with IL-1β, TNF-α, and IL-6 playing pivotal roles in amplifying and sustaining inflammatory signals. Specifically, IL-6 is a central mediator in the pathogenesis of chronic inflammatory disorders, autoimmune diseases, cancer, and cytokine storms [45,46]; our experimental data showed that COP supplementation significantly reduced serum IL-6 levels in Wenchang chicks, suggesting a direct inhibitory effect on this critical proinflammatory cytokine. Additionally, TNF-α is a pleiotropic cytokine that exerts diverse effects on various cell types, ranging from regulating cell proliferation and apoptosis to modulating immune responses. It is well-recognized as a master regulator of inflammatory reactions and has been implicated in the development of numerous inflammatory and autoimmune diseases—primarily by promoting the recruitment of inflammatory cells and the secretion of additional proinflammatory factors [47]. Consistent with its effect on IL-6, our study further confirmed that COP could decrease the serum expression level of TNF-α in Wenchang chicks. Taken together, these results indicate that COP may alleviate systemic inflammation in the organism by downregulating the expression of key proinflammatory cytokines (IL-6 and TNF-α), thereby supporting the maintenance of physiological homeostasis and overall health in Wenchang chicks.

The brooding period is a critical stage for chicks, characterized by an immature immune system and fragile physiological functions. The proper establishment and functional maturation of their immune competence serve as a core determinant for ensuring normal growth and development, reducing mortality rates, and optimizing breeding efficiency. Mounting evidence has highlighted the immunopotentiating effects of herbal medicines and their bioactive components [48,49]. Specifically, polysaccharides derived from medicinal herbs have been shown to exert multifaceted immunomodulatory effects: they can stimulate T-cell proliferation, upregulate the expression of cell surface antigens on lymphocytes, elevate serum antibody titers, and enhance the secretion of various cytokines [50,51,52]. Notably, changes in serum antibody agglutination titers provide an accurate and direct readout of humoral immune status, making it a reliable indicator for evaluating immune competence. In the present study, consistent with these findings, the COP-supplemented groups exhibited significantly higher antibody titers on days 7, 14, and 21 post the first vaccination. These results collectively demonstrate that dietary COP supplementation can enhance the resistance of chicks to Newcastle disease virus (NDV), thereby facilitating their successful transition through the vulnerable brooding period.

A growing body of research has demonstrated that the intestinal microbiota plays a crucial role in converting dietary components into bioactive metabolites, thereby directly or indirectly influencing intestinal barrier function and nutrient absorption [53,54]. Certain beneficial bacteria in the intestine, such as most Lactobacillus species and Clostridium butyricum, can promote the growth of intestinal epithelial cells, whereas some harmful bacteria (e.g., Proteobacteria) exert adverse effects on the gastrointestinal tract [55]. In the H-cop group, the dominant bacterial order *o__Oscillospirales* exhibited a high LDA score, indicating it served as the core dominant taxon in this group. Previous studies have confirmed that bacteria belonging to the order *Oscillospirales* possess dual functions: complex polysaccharide degradation and short-chain fatty acid (SCFA, particularly butyrate) synthesis [56,57]. Specifically, they secrete cellulases and hemicellulases to degrade dietary fiber that is indigestible by the host, converting it into SCFAs such as butyrate. As the primary energy source for intestinal epithelial cells (IECs), butyrate can promote IEC proliferation and the expression of tight junction proteins (occludin, ZO-1), directly strengthening the intestinal physical barrier. Meanwhile, butyrate can inhibit the nuclear factor-κB (NF-κB) signaling pathway by activating G protein-coupled receptors (GPR41/43), reducing the release of proinflammatory cytokines (e.g., TNF-α, IL-6) and alleviating intestinal chronic inflammation—findings that are consistent with the expression levels of tight junction proteins and inflammatory factors observed in this study. In the H-cop group, the genus *g__Lactobacillus* and its species *s__Lactobacillus* johnsonii also showed significant LDA advantages, with their functional effects focusing on intestinal pathogenic bacteria inhibition, immunomodulation, and barrier enhancement [58]. As a typical probiotic, *s__Lactobacillus* johnsonii can ferment carbohydrates to produce lactic acid, lowering the intestinal luminal pH to 4.5–5.5 and directly inhibiting the colonization of harmful bacteria (e.g., enteropathogenic E. coli, Salmonella) and the expression of their virulence genes [59]. Additionally, the bacteriocin it secretes (e.g., lactocin Johnsonii) can activate naive macrophages to differentiate into CD206^+^ macrophages and induce the release of IL-10 via the TLR1/2-STAT3 pathway [60], alleviating experimental colitis and further enhancing anti-inflammatory and probiotic effects. Meanwhile, this strain promotes the integrity of the barrier function in Caco-2 cells through GAPDH-JAM-2 binding [61]. At the immunomodulatory level, *g__Lactobacillus* can facilitate the differentiation of intestinal mucosal regulatory T cells (Tregs) [62] and enhance immune tolerance by secreting anti-inflammatory cytokines such as IL-10, thereby preventing intestinal damage caused by excessive immune activation. In the L-cop group, the dominant genera *g__Marvinbryantia* and *g__Sellimonas* are both butyrate-producing bacteria belonging to the phylum Firmicutes. Their differential enrichment between groups directly affects intestinal butyrate levels [63].

## 5. Conclusions

In summary, in this study, two polysaccharide fractions (designated as COP-1 and COP-2) were isolated and purified from the rhizomes of *Curculigo orchioides* using water extraction–ethanol precipitation combined with ultrasonic-assisted extraction. The monosaccharide composition and molecular weight of COP-1, the main fraction, were determined.

COP were demonstrated to enhance the anti-inflammatory capacity, antioxidant capacity, intestinal barrier function, and protein synthesis capacity of Wenchang female chicks. Regarding the mechanism underlying the promotion of growth and development in Wenchang chicks, it may involve regulating the intestinal microbiota and its metabolism, which in turn influences the expression of proteins related to the PI3K/Akt/mTOR pathway through microbial metabolites—ultimately promoting protein synthesis. For the enhancement of anti-inflammatory, antioxidant, and intestinal barrier functions, it is likely that dietary COP supplementation facilitates the enrichment of relevant bacterial species, thereby improving these physiological functions in Wenchang female chicks.

Limitations of this study include lack of in-depth analysis of COP-1’s chemical structure and insufficient exploration of the specific mechanisms underlying changes in microbial communities (e.g., Firmicutes, Bacteroidetes, and short-chain fatty acid (SCFA)-producing bacteria such as Oscillospirales) following COP treatment. Future research will investigate whether COP-1 promotes growth performance and protein synthesis in Wenchang pullets via gut microbiota through fecal microbiota transplantation experiments and in vitro simulation studies. This will further elucidate the mechanism of action of *C. orchioides* polysaccharides as a feed additive in the intestines of Wenchang pullets.

## Figures and Tables

**Figure 1 animals-15-03585-f001:**
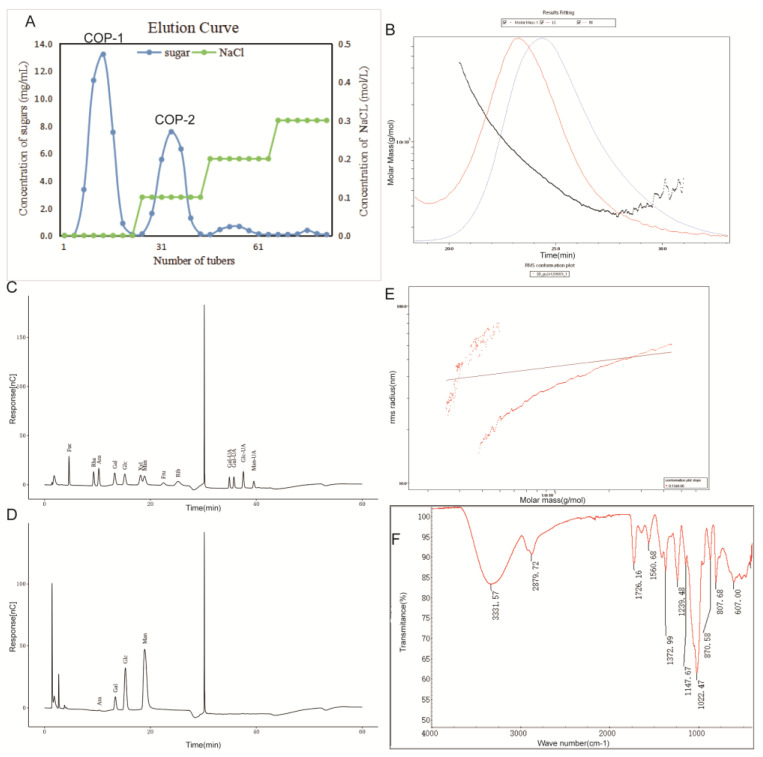
Isolation and purification of polysaccharides from *Curculigo orchioides*, molecular weight of the major component COP-1, monosaccharide composition of COP-1, and characterization of each component polysaccharide. (**A**) COP was eluted using 0, 0.1, 0.2, and 0.3 M NaCl solutions on a DEAE SepLife FF cellulose column, separating COP into two fractions: COP-1 and COP-2. (**B**) Molecular weight distribution of the major component COP-1. (**C**) Chromatogram of monosaccharide standards. (**D**) Chromatogram of the major component COP-1 after hydrolysis. (**E**) Molecular structure diagram of COP-1. (**F**) FT-IR spectrum of COP-1.

**Figure 2 animals-15-03585-f002:**
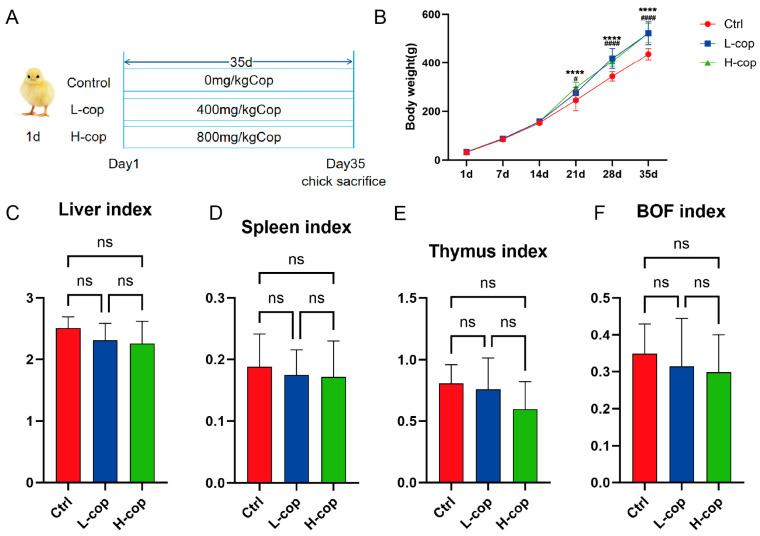
Effects of COP on growth performance of Wenchang female chicks. (**A**) Experimental flowchart. (**B**) Weight changes. (**C**) Liver index. (**D**) Spleen index. (**E**) Thymus Index. (**F**) Bursa of Fabricius index. Data are expressed as mean ± SEM and analyzed by one-way ANOVA (*n* = 8). ns *p* > 0.05, **** *p* < 0.0001: Control vs. H-cop; # *p* < 0.05, #### *p* < 0.0001: Control vs. L-cop.

**Figure 3 animals-15-03585-f003:**
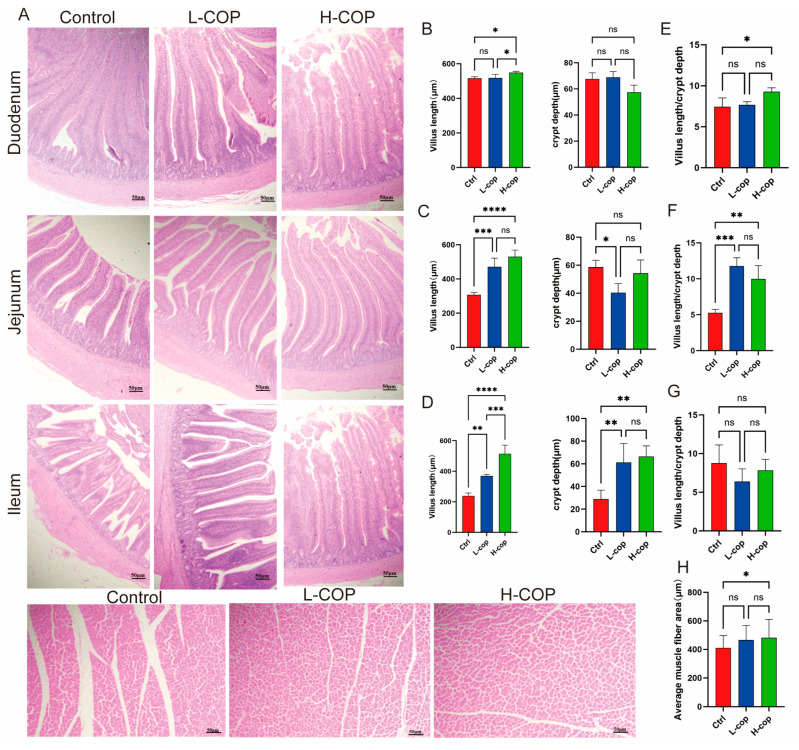
Effects of COP on H&E staining (40×) of intestinal morphology and muscle morphology in Wenchang chicks. (**A**) Microscope scanning. Villus length and crypt depth of duodenum (**B**), jejunum (**C**), and ileum (**D**) Villus length/crypt depth of jejunum (**E**), duodenum (**F**), and ileum (**G**). (**H**) Average muscle fiber area. Villus length: the linear distance from the tip of villi to the opening of intestinal gland. Crypt depth: the straight distance from the opening of crypt and the bottom of crypt. Abbreviations: H&E, hematoxylin and eosin; Data are expressed as mean ± SEM and analyzed by one-way ANOVA (*n* = 8). * *p* < 0.05, ** *p* < 0.01, *** *p* < 0.01, *****p* < 0.0001.

**Figure 4 animals-15-03585-f004:**
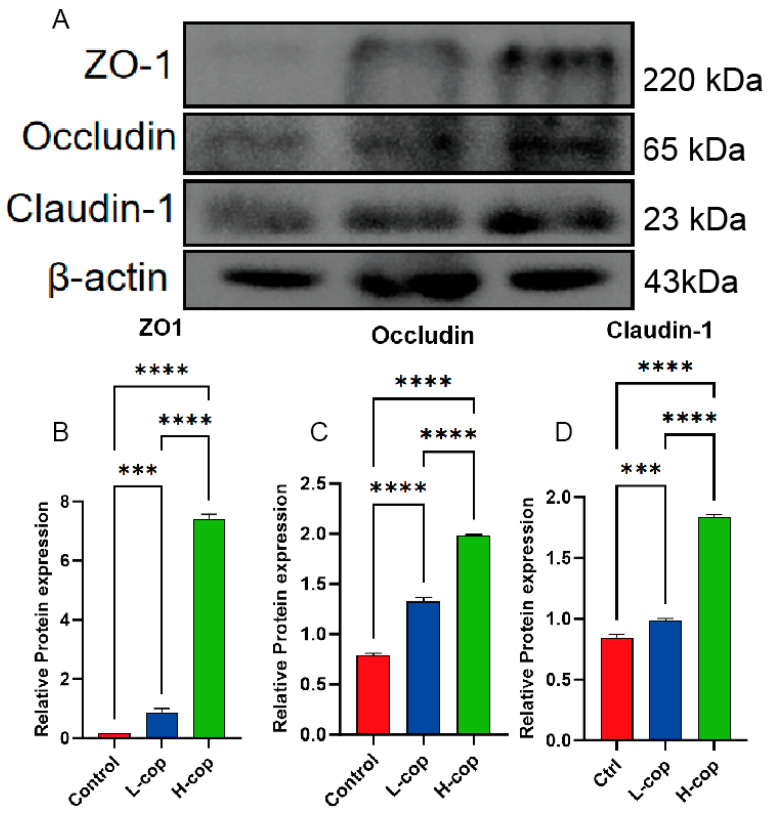
Effects of COP on tight junction protein expression in the colon of Wenchang female chicks. (**A**) Expression of tight junction proteins (ZO-1, occludin, and Claudin-1) in colon tissue. (**B**) Relative expression of ZO-1 protein. (**C**) Relative expression of occludin. (**D**) Relative expression of Claudin-1 protein. Data are presented as mean ± SEM and analyzed by one-way ANOVA (*n* = 8). *** *p* < 0.001, and **** *p* < 0.0001.

**Figure 5 animals-15-03585-f005:**
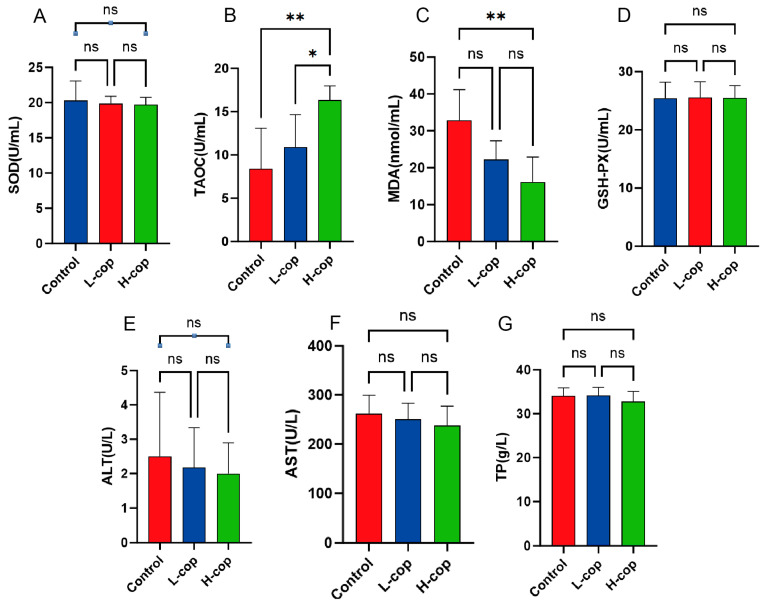
Effects of COP on antioxidant capacity and hepatic parameters in Wenchang chicks. (**A**–**D**) Serum levels of SOD, T-AOC, MDA, GSH-Px, and GSH. (**E**–**G**) Serum ALT, AST, and TP. Abbreviations: GSH-Px, glutathione peroxidase; MDA, malondialdehyde; SOD, superoxide dismutase; T-AOC, total antioxidant capacity; ALT, alanine aminotransferase; AST, aspartate aminotransferase. Data are presented as mean ± SEM and analyzed by one-way ANOVA (*n* = 8). ns *p* > 0.05, * *p* < 0.05, ** *p* < 0.01.

**Figure 6 animals-15-03585-f006:**
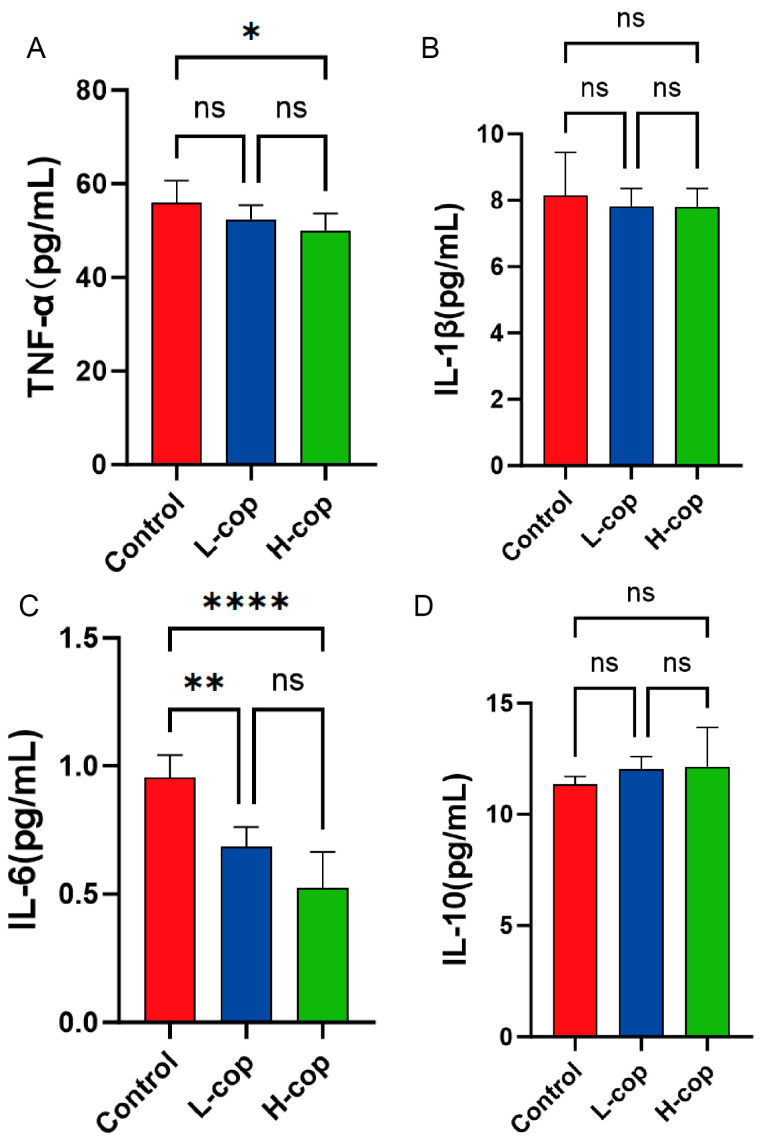
Effects of COP on serum levels of inflammation-related cytokines. (**A**) TNF-α content. (**B**) IL-1β content. (**C**) IL-6 content. (**D**) IL-10 content. Data are presented as mean ± SEM and analyzed by one-way ANOVA (*n* = 8). ns *p* > 0.05, * *p* < 0.05, ** *p* < 0.01, and **** *p* < 0.0001.

**Figure 7 animals-15-03585-f007:**
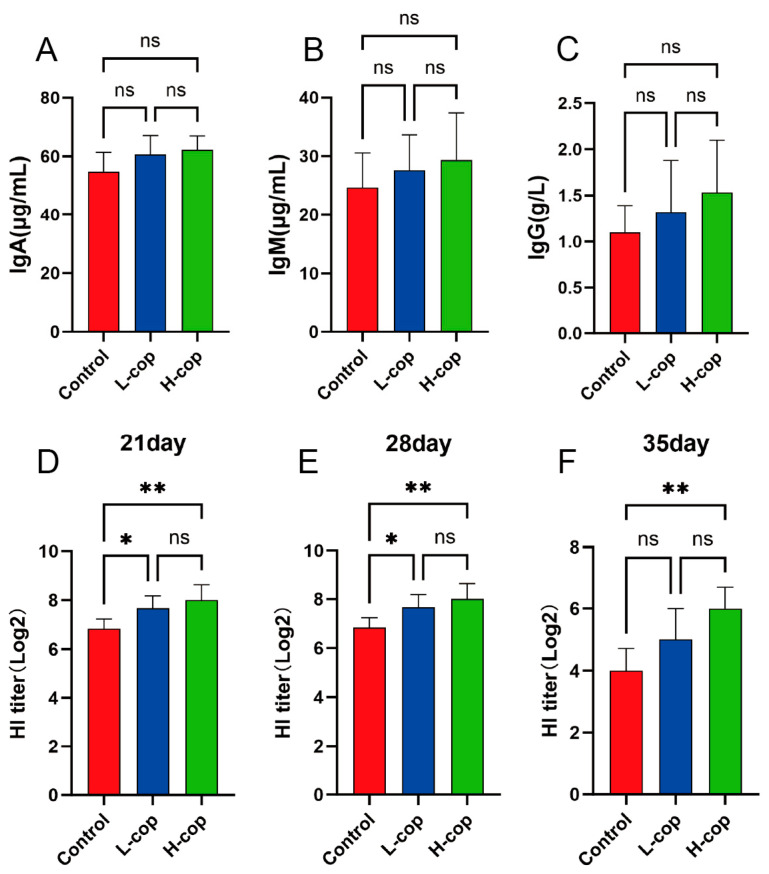
Effects of COP on serum immune-related parameters (**A**) Content of IgA; (**B**) Content of IgM; (**C**) Content of IgG; (**D**) Antibody titer on day 21; (**E**) Antibody titer on day 28; (**F**) Antibody titer on day 35. Data are presented as mean ± SEM and analyzed by one-way ANOVA (*n* = 8). ns *p* > 0.05, * *p* < 0.05, ** *p* < 0.01.

**Figure 8 animals-15-03585-f008:**
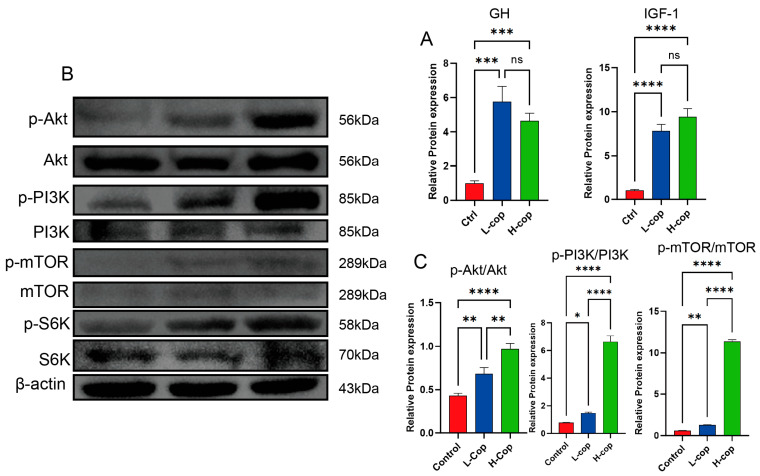
COP enhances growth factor expression and protein synthesis in Wenchang female chicks. (**A**) mRNA expression levels of growth hormone (GH) and insulin-like growth factor-1 (IGF-1); (**B**,**C**) Western blot analysis of the PI3K/Akt/mTOR signaling pathway. Data are presented as mean ± SEM and analyzed by one-way ANOVA (*n* = 8). ns *p* > 0.05, * *p* < 0.05, ** *p* < 0.01, *** *p* < 0.001, and **** *p* < 0.0001.

**Figure 9 animals-15-03585-f009:**
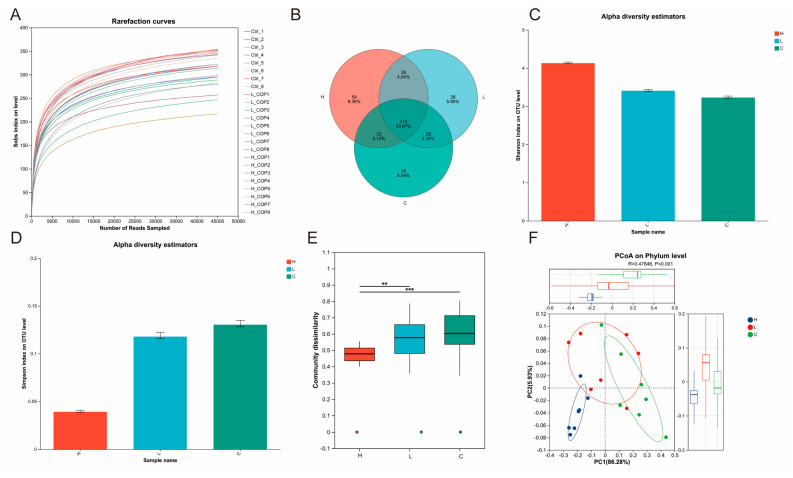
Effects of COP on the cecal microbiota of Wenchang female chicks (*n* = 8). (**A**) Dilution curve. (**B**) OTU overlap Venn diagram. (**C**) Simpson index. (**D**) Shannon index. (**E**) Species β-diversity. (**F**) Analysis of gut microbial structure using principal component analysis (PCoA). ** *p* < 0.01, *** *p* < 0.001.

**Figure 10 animals-15-03585-f010:**
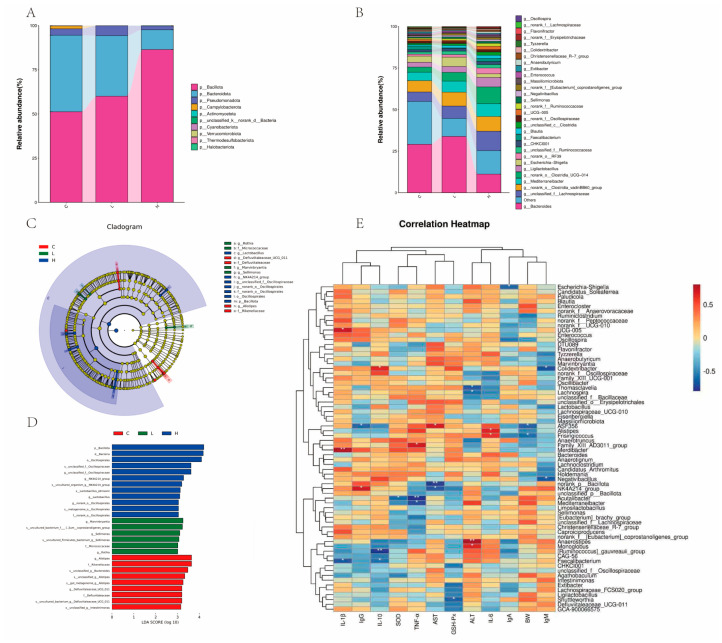
(**A**) Relative abundance of microbial composition at the phylum level; (**B**) Relative abundance of microbial composition at the genus level; (**C**) Linear Discriminant Analysis Effect Size (LEfSe) plot identifying the taxa with the most significant differences in abundance among each group; (**D**) Analysis of differences in microbial taxa among the four groups using Linear Discriminant Analysis (LDA) effect size; (**E**) Spearman correlation analysis between different indicators and the top 69 genera (*n* = 8). Row names represent different indicators, and column names represent phyla and genera. Orange and blue squares represent positive and negative correlations, respectively, with the intensity of the color indicating the degree of correlation. * *p* < 0.05, ** *p* < 0.01.

**Table 1 animals-15-03585-t001:** Oligo nucleotide primers in Wenchang chicks.

Genes	Sequences (5′ → 3′)	Product Size (bp)	Accession No.
growth hormone 1	F:GCTGCCGAGACATATAAAGAGT	109	NM_204359.2
	R:GAGCTGGGATGGTTTCTGAGT		
insulin-like growth factor 1	F:TGTGCTCCAATAAAGCCACCT	117	NM_001004384.3
	R:TCCTGTGTTCCCTCTACTTGT		
GAPDH	F:GTCTGGAGAAACCAGCCAAGTA	149	NM_204305.2
	R:CGCATCAAAGGTGGAGGAATG		

**Table 2 animals-15-03585-t002:** Monosaccharide composition of COP-1.

Monosaccharide	Molar Ratio
Ara	0.32
Gal	6.34
Glc	24.38
Man	68.97

**Table 3 animals-15-03585-t003:** Gel Permeation Chromatography Analysis of COP.

	Mn	Mp	Mw	Mz	Mw/Mn
(kDa)	43.727	51.773	58.822	92.751	2.121

**Table 4 animals-15-03585-t004:** Effects of COP on Growth Performance of Wenchang chicks.

Dietary COP Level				
Items	Control	L-Cop	H-Cop	*p*-Value
BW	436.35 ± 23.69 ^b^	524.45 ± 39.10 ^a^	522.6 ± 47.10 ^a^	<0.001
FI	147.59 ± 1.18 ^b^	141.04 ± 1.07 ^c^	158.70 ± 0.64 ^a^	<0.001
FCR	2.57 ± 0.16 ^a^	2.03 ± 0.20 ^c^	2.38 ± 0.18 ^b^	<0.001

Data are expressed as mean ± SEM and analyzed by one-way ANOVA, a–c Means with different superscripts within the same column differ significantly (*p* < 0.05).

## Data Availability

The data that support the findings of this study are available from the corresponding authors upon reasonable request.

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
