# Peer review of "Curculigo orchioides Polysaccharide Promotes the Growth and Development of Wenchang Chickens via the PI3K/Akt/mTOR Signaling Pathway"

_animals, 2025, doi:10.3390/ani15243585_

Round 1

Reviewer 1 Report

Comments and Suggestions for Authors

Attached

Comments on the Quality of English Language

Can be improved. 

Author Response

Abstract:  

Comment 1:Please define COP before using the abbreviation.
Response: Thank you for pointing this out . I agree with this perspective, therefore, I have defined COP as Curculigo orchioides Polysaccharide in the abstract

Comment 2:1-35-day old is confusing. I’d rather suggest 1 to 35 days old. .
Response: Thank you for your correction. I also agree with your point of view, teacher. Therefore, I have revised the abstract as per your requirements

Comment 3:‘against Newcastle disease virus’ is repeated twice in the sentence unnecessarily.
Response: Thank you for your correction, teacher. I also agree with your point of view, so I have deleted the redundant expressions in the abstract

Comment 4: The abbreviations such as FI, FCR, T-AOC, MDA, TNF-α, IL-6, NDV are unnecessary in the abstract as they are not 17 repeated (a suggestion to save words).
Response:Thank you for your correction, teacher. I also agree with your point of view. Therefore, I have removed the abbreviations in the abstract to make the expression more concise

Introduction: 

Comment 1: 3rd paragraph: Please introduce MC3T3-E1 cells.
Response:Thank you, teacher, for your correction. It was my oversight when writing the article. Therefore, I have already introduced the MC3T3-E1 cells in the introduction

Comment 2: End of last paragraph: The objective does not reflect the title (Growth and Development of Wenchang Chicken). Please  rephrase. 
Response:Thank you, teacher, for your correction. It was my oversight when writing the article. Therefore, I have resummarized the last paragraph of the introduction to make the objective consistent with the title

Materials and Methods: 

Comment 1:The materials section alone is unnecessary. It can be merged with other sections’ i.e. supplies (antibodies) for western blot and ELISA could be mentioned under those specific sections.
Response: Thank you for pointing this out . I agree with this perspective, Therefore, I have cut out the material section and made the content of the material section reflected in the corresponding chapters, such as the ELIZA chapter and the WB chapter

Comment 2:
 The abstract mentions 120 total birds in three treatment groups, with 5 replicates per group and 8 chickens per replicate. However, the ‘animal experiment’ section mentions 120 total birds 3 experimental groups with 10 chicks per replicate and 8 replicates per group (this brings the total to 240). Please correct this. 
Response: Thank you for your correction. This was a very serious oversight. I have already corrected this mistake in the corresponding part. I'm sorry again

Comment 3:Mention how many days the chicks were alive until the end of the experiment. 
Response: Thank you for your correction. There was a problem with my expression when writing the article, which led you, teacher, to fail to notice my experimental period. I have already added the age of the chicks at the end of the experiment in the animal experiment section

Comment 4:Growth performance section: rewrite the ‘Calculate the weight gain (BWG) and FCR……….’ sentence in passive voice
Response: Thank you for your correction, teacher. I fully agree with your point of view, so I have already changed the grammar to the passive voice in the corresponding part.

Comment 5:Serum biochemical detection: mention the source (Company name and location) of the Hitachi 3500 automatic  
biochemical analyzer. 
Response:Thank you, teacher, for your correction. This is because I was negligent when writing articles and didn't complete all the information. Therefore, I have supplemented the corresponding information

Comment 6:qRT-PCR section: What was the spectrophotometer? (model and location)  
Response:Thank you, teacher, for your correction. This is because I was negligent when writing articles and didn't complete all the information. Therefore, I have supplemented the corresponding information.

Comment 7:Western blot detection of related protein expression section: Mention the models and company locations of the protein extraction kit and the BCA protein assay kit. 
Response:Thank you, teacher, for your correction. This is due to my negligence when writing the article and I didn't complete the information completely. Therefore, I have supplemented the origin information of the corresponding protein extraction kit and BCA protein determination kit.

Comment 8:Same as above for the ELISA kit.
Response:Thank you for your correction, teacher. This is because I was negligent when writing articles and didn't complete all the information. Therefore, I have supplemented the origin information of the ELISA determination kit.

Results section:  

Comment 1: Move this to the materials and methods section: During COP extraction, accurately weigh 1500 grams of Curculigo orchioides. Employ ultrasonic-assisted extraction and ethanol static precipitation to extract COP, followed by freeze drying. A total of 100 grams of COP was obtained, yielding a crude extraction rate of 6.6%. For primary purification,  precisely weigh 30 grams of COP. Add sufficient water to completely dissolve the crude polysaccharides, then add 0.4– 0.6 grams each of papain and complex protease for overnight enzymatic hydrolysis. Subsequently, remove proteins 
 using Sevage reagent and lipids using petroleum ether. Further purify by re-moving polysaccharide impurities through 
 macroporous resin AB-8 and a 3000-Dalton dialysis bag. Finally, COP was purified using a DEAE Sepharose FF 
 cellulose column. The procedure involved loading 5.0 g of COP as the sample. After elution with a gradi-ent of NaCl 
 solutions, COP separated into two fractions (COP-1 and COP-2) (Figure 1A). As these fractions constituted the 
 majority of COP, they were collected and freeze-dried. Following lyophilization, the mass and purity of the two major 
 fractions were deter-mined: Also, make sure that all sentences are in passive voice. 
Response:Thank you, teacher, for your correction. I completely agree with your point of view. Therefore, I have placed the corresponding parts in Materials and Methods and changed all the sentences to the passive voice

Comment 2: The caption of table 3 is bigger and looks like an image. Please correct it.  
Response:Thank you, teacher, for your correction. I fully agree with your point of view, so I have already reduced the font size of the title in Table 3. Finally, I would like to express my sincere gratitude once again to you for taking the time out of your busy schedule to comment on my article and offer very important suggestions. I am extremely grateful to you, teacher. I wish you good health and further progress in your career

Reviewer 2 Report

Comments and Suggestions for Authors

This study explored the potential application of COP as feed additives, which offers a promising direction for healthy poultry farming. The research investigated multiple levels from growth performance, antioxidant capacity, and immunity to intestinal health and molecular mechanisms. The data are comprehensive. But there is room for improvement in terms of methodological rigor, depth of result interpretation, and mechanistic elucidation.

Major Concerns and Suggestions for Revision:

  1. Material and Methods

Insufficient polysaccharide purity: The manuscript mentions that the purity of COP-1 was 84.1%, but the method for determining this purity is not specified. It is recommended to supplement more detailed for determining this purity.

Ambiguity in animal experimental details: The manuscript states "10 chicks per replicate and 8 replicates per group," but in the subsequent statistical analysis, it mentions "n=8," which is contradictory. Please clarify the final sample size (n) used for analysis. Was it one chicken per replicate (totaling 8), or was another method used? A clear definition of sample size is crucial for statistical power.

Some reagents only provide the company name without the city and country.

  1. Results

Correlation vs. Causation in gut microbiota: The study found that COP altered the gut microbiota composition, enriching beneficial bacteria. The authors speculate in the discussion that microbial metabolites (e.g., butyrate) might mediate these effects. However, this remains a correlational inference. To strengthen the argument, the discussion could more explicitly state that this is a direction for future validation, for instance, by measuring short-chain fatty acid (SCFA) concentrations in cecal contents or conducting fecal microbiota transplantation experiments to directly verify causality.

Upstream regulation of the PI3K/Akt/mTOR Pathway: The study confirms that COP activates the PI3K/Akt/mTOR pathway and detects increased mRNA expression of GH-1 and IGF-1. However, how COP leads to the upregulation of GH-1 and IGF-1. Is it through direct action on endocrine organs, or indirectly by improving intestinal health, which in turn affects nutrient absorption and hormone secretion? Exploring these upstream events would provide a more complete mechanistic picture.

  1. Figures

Presentation of Figure 8: Figure 8A shows the mRNA expression of GH and IGF-1, but the results of this part showed that “COP supplementation increased the mRNA expression levels of the two growth factors (Fig. 8B)”, which can be confusing. It is recommended to unify the figure labeling to ensure clear correspondence.

Data consistency: The "Materials and Methods" section mentions "10 chicks per replicate and 8 replicates per group," but the analysis of gut microbiota specifies "n=7". Please verify and standardize the description of sample size throughout the manuscript to avoid confusing readers.

  1. Discussion:

The logical connection between some paragraphs in the discussion could be tighter. For example, after discussing the gut microbiota, it could be more explicitly linked to the previously mentioned antioxidant, anti-inflammatory, and intestinal barrier functions. This would create a more integrated narrative of "COP-Microbiota-Metabolites-Host Health" rather than presenting them as relatively independent points.

Recommendation: Minor Revision. I hope the authors will carefully consider the comments above, especially the supplementation of methodological details and the deepening of the mechanistic discussion, which will further enhance the scientific value of the paper.

  1. References

The format of references in the manuscript is not consistent, please check the entire manuscript thoroughly and modify each item according to the requirements of the magazine. Moreover, in the references section, No. 17 is an article from a Chinese journal, and didn’t cited in the manuscript.

Author Response

Material and Methods

Comment 1:Insufficient polysaccharide purity: The manuscript mentions that the purity of COP-1 was 84.1%, but the method for determining this purity is not specified. It is recommended to supplement more detailed for determining this purity.
Response: Thank you for pointing this out . I agree with this perspective, Due to technical and material limitations, the polysaccharide concentration did not reach over 95%. I'm very sorry. At that time, when I reviewed the literature, I found that the polysaccharide concentrations in some papers [1][2] were similar to mine. Therefore, I directly used 84.1% polysaccharide concentration for the subsequent experiments
[1]Bing Yang, Xiaofeng Li, Noura M. Mesalam, Mohamed Farouk Elsadek,  Abdel-Moneim Eid Abdel-Moneim,The impact of dietary supplementation of polysaccharide derived from Polygonatum sibiricum  on growth, antioxidant capacity, meat quality, digestive physiology,  and gut microbiota in broiler chickens,Poultry Science,Volume 103, Issue 6,2024
103675, ISSN 0032-5791, https://doi.org/10.1016/j.psj.2024.103675.
[2]Yingying Qiao, Changzhong Liu, Yongpeng Guo, Wei Zhang, Weibing Guo, Kyselov Oleksandr,  Zhixiang Wang,Polysaccharides derived from Astragalus membranaceus and Glycyrrhiza uralensis improve growth performance  of broilers by enhancing intestinal health and modulating gut microbiota,Poultry Science,Volume 101,  Issue 7202, 2101, 905, ISSN 0032-5791, https://doi.org/10.1016/j.psj.2022.101905.

Comment 2:Ambiguity in animal experimental details: The manuscript states "10 chicks per replicate and 8 replicates per group," but in the subsequent statistical analysis, it mentions "n=8," which is contradictory. Please clarify the final sample size (n) used for analysis. Was it one chicken per replicate (totaling 8), or was another method used? A clear definition of sample size is crucial for statistical power.
Response: Thank you for pointing this out . I agree with this perspective, I'm very sorry for making such a big mistake when writing the thesis. I have already made corrections in the corresponding positions. Thank you, teacher, for your criticism and correction

Comment 3:Some reagents only provide the company name without the city and country.
Response: Thank you for pointing this out . I agree with this perspective, I apologize again for this oversight. I have already provided the specific information of the manufacturers regarding the relevant reagents and consumables.

Results:

Comment 1:Correlation vs. Causation in gut microbiota: The study found that COP altered the gut microbiota composition, enriching beneficial bacteria. The authors speculate in the discussion that microbial metabolites (e.g., butyrate) might mediate these effects. However, this remains a correlational inference. To strengthen the argument, the discussion could more explicitly state that this is a direction for future validation, for instance, by measuring short-chain fatty acid (SCFA) concentrations in cecal contents or conducting fecal microbiota transplantation experiments to directly verify causality.
Response: Thank you for pointing this out . I agree with this perspective, Therefore, I have already pointed out in the discussion section that the main verification direction of future experiments is whether COP mediates the growth performance of chicks through intestinal microbiota. Thank you, teacher, for your guidance, which provides a clear idea for future experiments.

Comment 2:Upstream regulation of the PI3K/Akt/mTOR Pathway: The study confirms that COP activates the PI3K/Akt/mTOR pathway and detects increased mRNA expression of GH-1 and IGF-1. However, how COP leads to the upregulation of GH-1 and IGF-1. Is it through direct action on endocrine organs, or indirectly by improving intestinal health, which in turn affects nutrient absorption and hormone secretion? Exploring these upstream events would provide a more complete mechanistic picture.
Response:Thank you, teacher, for your suggestion. From the current experiments, COP stimulates growth factors through the mediation of intestinal microbiota. Subsequent experiments will verify this view through in vitro experiments to make my experiments more rigorous and standardized. Thank you, teacher, for your guidance

Figures:

Comment 1:Presentation of Figure 8: Figure 8A shows the mRNA expression of GH and IGF-1, but the results of this part showed that “COP supplementation increased the mRNA expression levels of the two growth factors (Fig. 8B)”, which can be confusing. It is recommended to unify the figure labeling to ensure clear correspondence.
Response:Thank you for your criticism, teacher. I have corrected the relevant mistakes in the article. I'm very sorry that such a mistake occurred in the article. I have unified the numerical tags of the article and will definitely not make such a mistake again.

Comment 2:Data consistency: The "Materials and Methods" section mentions "10 chicks per replicate and 8 replicates per group," but the analysis of gut microbiota specifies "n=7". Please verify and standardize the description of sample size throughout the manuscript to avoid confusing readers.
Response:Thank you, teacher, for your criticism. I have corrected the error of inconsistent sample sizes in the article. I'm very sorry that such a mistake occurred in the article.

Discussion:

Comment 1:The logical connection between some paragraphs in the discussion could be tighter. For example, after discussing the gut microbiota, it could be more explicitly linked to the previously mentioned antioxidant, anti-inflammatory, and intestinal barrier functions. This would create a more integrated narrative of "COP-Microbiota-Metabolites-Host Health" rather than presenting them as relatively independent points.
Response:Thank you for your correction, teacher. I fully agree with your viewpoint. Therefore, I have already written in the discussion part of the article about the connection between the intestinal microbiota and antioxidant, anti-inflammatory and intestinal barrier functions.

References:

Comment 1:The format of references in the manuscript is not consistent, please check the entire manuscript thoroughly and modify each item according to the requirements of the magazine. Moreover, in the references section, No. 17 is an article from a Chinese journal, and didn’t cited in the manuscript.
Response:Thank you for your criticism, teacher. I fully agree with your point of view. Previously, when I was checking the citation format on the official website, I saw the APA format, so I directly used the APA citation format for the article at that time. Now, I have completely changed it to the numeric format to make the article more standardized. I typed the names of the references in English at that time, which might have led you, teacher, to think that I didn't quote them. In fact, I have already quoted them. Now I have modified the Chinese in the references to the English format.

Reviewer 3 Report

Comments and Suggestions for Authors

This study provides a systematic evaluation of the effects of Curculigo orchioides polysaccharides on the growth and development of Wenchang Chicken. The research is well-structured, with a clearly articulated foundation in the introduction and effectively presented results. The introduction successfully establishes the background, scientific significance, and specific objectives of the study, while the results section offers intuitive figures and substantial information, making it a valuable reference for related fields. However, there is room for improvement in terms of academic rigor and methodological reproducibility. The following suggestions are offered to enhance the quality of the manuscript:

1 The abstract should be expanded to include a clearer logical connection between the core mechanisms, particularly the interaction between gut microbiota changes and relevant metabolic pathways, thereby aligning with the mechanistic hypotheses proposed in the conclusion.

2 The introduction section supplements the background of cross-regulation to reinforce the rationality of the research mechanism.

3 The methods section should include more detailed descriptions of the rearing environment parameters and conditions, such as temperature, humidity, light cycles, and stocking density, to improve experimental reproducibility.

4 Literature citations need to adhere to academic standards. For instance, Chinese references such as Bai et al. (2024) should be supplemented with corresponding English translations to meet international academic exchange and indexing requirements.

Author Response

Comment 1:The abstract should be expanded to include a clearer logical connection between the core mechanisms, particularly the interaction between gut microbiota changes and relevant metabolic pathways, thereby aligning with the mechanistic hypotheses proposed in the conclusion.
Response:Thank you for your criticism, teacher. I fully agree with your viewpoint. Therefore, I have expanded on the application of the abstract to clarify the connection between the gut microbiota and pathways, echoing the speculation in the conclusion.

Comment 2:The introduction section supplements the background of cross-regulation to reinforce the rationality of the research mechanism.
Response:Thank you for your criticism, teacher. I fully agree with your point of view. Therefore, I have already supplemented the background of cross-regulation in the introduction, strengthening the rationality of the research mechanism.

Comment 3: The methods section should include more detailed descriptions of the rearing environment parameters and conditions, such as temperature, humidity, light cycles, and stocking density, to improve experimental reproducibility.
Response:Thank you for your criticism, teacher. I fully agree with your point of view. Therefore, I have already supplemented the breeding details in the experimental design part to make the experiment easier to reproduce

Comment 4: Literature citations need to adhere to academic standards. For instance, Chinese references such as Bai et al. (2024) should be supplemented with corresponding English translations to meet international academic exchange and indexing requirements.
Response:Thank you for your criticism, teacher. I fully agree with your point of view. Therefore, I have already supplemented the breeding details in the experimental design part to make the experiment easier to reproduce

Round 2

Reviewer 2 Report

Comments and Suggestions for Authors The author has made modifications based on my previous suggestions. The manuscript has been sufficiently improved.